# Perceptual Neural Video Compression with Video Variational AutoEncoder at Low Bitrates

## Abstract

Existing neural video compression methods typically rely on frame-wise coding frameworks. Motion estimation and compensation are used to eliminate inter-frame redundancy, and compression performance is further enhanced through explicit residual or implicit conditional coding. However, these methods are primarily optimized for distortion, leading to significant degradation in perceptual quality at low bitrates. In this paper, we propose a novel learning-based video compression framework that leverages the compression and generative capabilities of video variational autoencoders. Unlike traditional frame-wise processing, our method operates on groups of frames, effectively improving perceptual quality at low bitrates. Specifically, we utilize video variational autoencoders to eliminate both temporal and spatial redundancy, encoding video clips into a perception-oriented latent space. Then, transform coding is applied to further capture spatial dependencies, yielding a more compressible latent representation. Finally, entropy coding is used to compress the quantized latent representation of the group of frames. Since each group of pictures is treated independently, our method can naturally be processed in parallel for acceleration. To incorporate information from adjacent frame groups and maintain temporal consistency across groups, we introduce an overlapping processing strategy, ensuring smooth transitions between adjacent groups. Extensive experimental results on benchmark datasets such as HEVC, UVG, and MCL-JCV demonstrate that our framework outperforms existing methods in terms of perceptual metrics.

## 1 Introduction

With the widespread application of mobile communications, satellite data transmission, and Internet of Things (IoT), the demand for efficient video compression under extreme bandwidth constraints has become increasingly critical (Cisco, 2017; Ericsson, 2024). Although existing coding methods can achieve acceptable pixel-wise distortion metrics in such scenarios, they often suffer from a substantial degradation in perceptual quality. Traditional approaches (Sullivan et al., 2012; Bross et al., 2021) exhibit detail loss due to block artifacts and motion distortions, whereas deep learning-based neural video codecs (NVCs) (Lu et al., 2019; Lombardo et al., 2019; Habibian et al., 2019; Pessoa et al., 2020; Hu et al., 2021; Li et al., 2021; 2023a; 2024b) have improved compression performance but primarily optimize for mean squared error (MSE), leading to overly smooth reconstructions. Under extreme bandwidth constraints, minimizing distortion exclusively becomes insufficient; instead, greater emphasis should be placed on perceptual quality and the preservation of fine textures to enhance the human visual experience (Blau & Michaeli, 2019).

To enhance perceptual quality in neural video compression, researchers have explored generative models to refine the framework. PLVC (Yang et al., 2022) introduces a recurrent conditional GAN, and HVFVC (Li et al., 2023b) employs confidence-based feature reconstruction along with periodic compensation loss. However, these methods primarily target high bitrates ($> 0.03$ bpp), where both traditional and learning-based video codecs already achieve visually tolerable results. In contrast, severe perceptual degradation becomes increasingly evident at lower bitrates ($< 0.03$ bpp), highlighting the very issue that our work seeks to resolve, as illustrated in Figure 1.

Figure 1: Comparison of reconstructed frames from different methods. Our approach preserves finer details and textures. Zoom in for better visibility of texture details.

A potential factor that could contribute to the difficulty of achieving lower bitrates with good perceptual quality in existing video compression methods may lie in their reliance on a frame-by-frame coding paradigm. Both traditional codec standards (Sullivan et al., 2012; Bross et al., 2021) and deep learning methods (Lu et al., 2019; Hu et al., 2021; Li et al., 2021; 2023a; 2024b; Jia et al., 2025b) adopt motion-based coding architectures, processing video frames sequentially. These approaches rely on motion estimation and compensation to reduce inter-frame redundancy, further improving coding efficiency either by explicitly compressing residual signals or by implicitly modeling feature-domain correlations through conditional coding. However, at low bitrates, this paradigm faces two fundamental bottlenecks. First, it is prone to cumulative error propagation (Lu et al., 2020; Li et al., 2024b). The independent quantization of motion and residual information leads to step-by-step error amplification, resulting in suboptimal bitrate allocation. Second, its temporal modeling capability is inherently limited (Hu et al., 2021; Sheng et al., 2024). Single-frame or short-term predictions struggle to capture long-range temporal dependencies, leading to temporal incoherence and increased bitrate consumption. To enhance perceptual quality at low bitrates, it is crucial to move beyond frame-by-frame prediction and incorporate global spatio-temporal modeling.

Recent advances in video generation offer a promising solution to the aforementioned bottlenecks. Video generation models typically employ video Variational AutoEncoders (VAEs) to capture both spatial and temporal dependencies by processing video clip-by-clip. Although video VAE-based frameworks such as Sora (Brooks et al., 2024) and VideoCrafter (Chen et al., 2024a) have achieved impressive results, their potential for video compression remains largely underexplored. Video VAEs encode video clips into compact latent variables through spatio-temporal joint encoding, enabling long-range redundancy elimination. However, applying video VAEs to compression presents several key challenges, including limited control over the rate-perception trade-off, insufficient compression ratios for low-bitrate scenarios, and the potential for discontinuities between video clips due to independent clip processing and quantization at low bitrates.

To address these challenges, this paper introduces a generative video compression framework based on video VAE as illustrated in Figure 2. Unlike conventional coding paradigms that rely on complex motion estimation, compensation, along with residual or conditional coding modules, our approach encodes an entire group of pictures (GoP) in an end-to-end manner. By leveraging the spatio-temporal joint modeling capability of video VAE, the framework effectively eliminates both temporal and spatial redundancy, which enables efficient video compression at low bitrates. To further reduce spatial redundancy, the latent variables produced by video VAE undergo an analysis transform and are subsequently compressed into a compact bitstream via entropy coding. During decoding, the model utilizes reconstruction priors embedded in video VAE to restore video details from the compressed latent representation, mitigating the high-frequency information loss caused by quantization. Additionally, to ensure inter-GoP continuity, adjacent GoPs are processed using an overlapping coding strategy. Our key contributions are summarized as follows:

- We propose a novel learning-based video compression framework that eliminates the need for motion estimation and motion compensation by leveraging video VAE to model temporal dependencies.

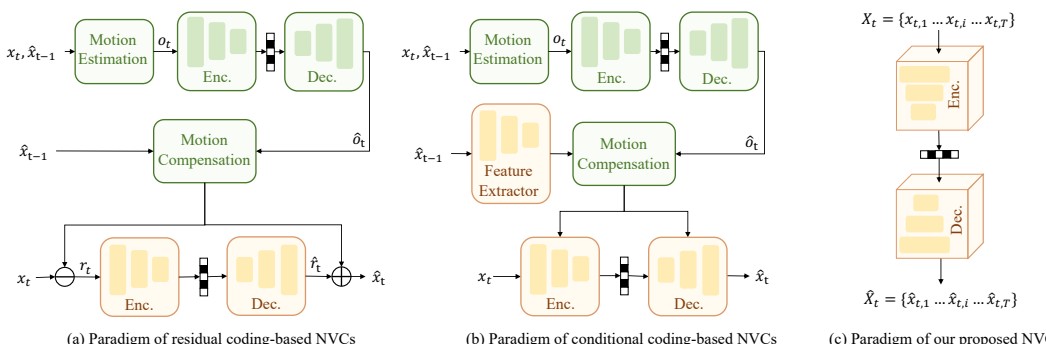

(a) Paradigm of residual coding-based NVCs   (b) Paradigm of conditional coding-based NVCs   (c) Paradigm of our proposed NVC

Figure 2: Paradigm shift in our proposed method. Instead of frame-wise processing, we adopt a group-by-group approach. Each video clip is encoded using a video variational autoencoder, which eliminates both temporal and spatial redundancy, mapping it into a perceptually optimized latent space to maintain visual quality at low bitrates.

- We optimize for perceptual quality at low bitrates, effectively addressing the severe visual degradation commonly observed in extreme compression scenarios.

- We introduce a training-free strategy that ensures temporal continuity across different GoPs, enhancing overall video consistency.

## 2  RELATED WORK

### 2.1  DISTORTION-ORIENTED COMPRESSION

The objective of distortion-oriented compression is to minimize bitrate by eliminating redundancy while preserving pixel-level fidelity. This goal has been extensively studied in image and video compression, with existing methods broadly categorized into traditional and learning-based codecs.

In image compression, traditional codecs (Wallace, 1991; Skodras et al., 2001; Bellard, 2018) rely on manually designed modules, such as the discrete cosine transform (DCT), to improve compression efficiency based on the rate-distortion principle. In contrast, learning-based codecs (Ballé et al., 2018; Minnen et al., 2018; He et al., 2021; Zhu et al., 2022) leverage image VAEs for feature extraction and utilize hyperprior models to further eliminate spatial redundancy. Ballé et al. (2018) first introduce the use of hyperprior information to model the distribution of latent representations for entropy coding. Subsequent advancements have surpassed traditional methods by optimizing network architectures (Liu et al., 2023; Li et al., 2024a) and refining entropy models (He et al., 2021; 2022; Qian et al., 2022; Jiang & Wang, 2023).

In video compression, traditional codecs (Bross et al., 2021; Sullivan et al., 2012) typically adopt a hybrid coding framework to eliminate spatial and temporal redundancy through intra-frame and inter-frame prediction. Learning-based codecs (Lu et al., 2019; Hu et al., 2021; Li et al., 2021; 2023a; 2024b; Jia et al., 2025a) have surpassed traditional methods in compression performance. Inspired by the hybrid framework, DVC (Lu et al., 2019) introduces the first end-to-end optimized video compression model, replacing key components of H.264/H.265 with deep neural networks. FVC (Hu et al., 2021) operates in feature space, improving efficiency. DCVC (Li et al., 2021) replaces explicit residual coding with conditional coding for better redundancy elimination, and DCVC-FM (Li et al., 2024b) further enhances performance with learnable quantization scalers and a periodically refreshing mechanism.

However, existing NVCs remain constrained by the conventional motion-based coding paradigm, which relies on frame-by-frame processing and employs image VAEs to compress residual and motion information. As a result, these approaches primarily focus on motion modeling (e.g., optical flow) and temporal redundancy elimination. Since the compressed content is limited to motion and residuals, they struggle to fully exploit the advanced spatial compression techniques developed in image compression.

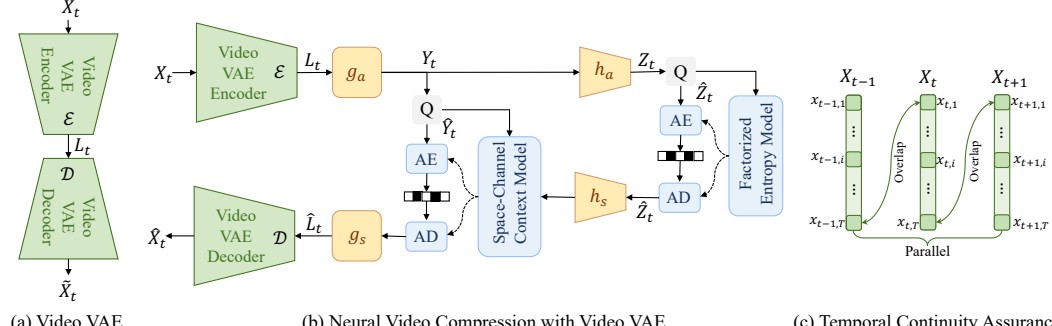

(a) Video VAE      (b) Neural Video Compression with Video VAE      (c) Temporal Continuity Assurance

Figure 3: Illustration of our proposed method. (a) The video variational autoencoder encodes the video clip $X_t$ into a perception-oriented latent variable $L_t$. (b) Transform coding is applied to $L_t$, producing a more compressible latent representation $Y_t$ for improved efficiency. AE and AD denote arithmetic encoding and decoding, while Q represents scalar quantization. (c) An overlapping processing strategy is introduced to enhance temporal consistency.

## 2.2 PERCEPTION-ORIENTED COMPRESSION

Most existing learning-based approaches are optimized for rate-distortion, which often leads to noticeable artifacts at low bitrates, significantly degrading perceptual quality. To mitigate this issue, recent studies have introduced perceptual loss optimization and generative models. These approaches permit imperceptible distortions while preserving perceptual quality and achieving better bitrate efficiency.

In image compression, HiFiC (Mentzer et al., 2020) introduces generative adversarial networks (GANs) (Goodfellow et al., 2014) for perceptual optimization, while MS-ILLM (Muckley et al., 2023) proposes a non-binary discriminator to further enhance visual quality. With the rise of diffusion models (Ho et al., 2020), CDC (Yang & Mandt, 2023) leverages a diffusion-based decoder, conditioning the diffusion process on compressed latents. PerCo (Careil et al., 2023) utilizes a pretrained text-to-image latent diffusion model to achieve perception-oriented compression at ultra-low bitrates. GLC (Jia et al., 2024) proposes a categorical hyperprior to compress the latent representation of a VQVAE (Esser et al., 2021).

In video compression, PLVC (Yang et al., 2022) introduces a recurrent conditional discriminator to enhance perceptual quality metrics beyond HEVC. HVFVC (Li et al., 2023b) surpasses VVC by leveraging confidence-based feature reconstruction and periodic compensation loss. However, these methods primarily target relatively high bitrates ($> 0.03$bpp), where visual degradation is minimal. In contrast, severe perceptual degradation becomes evident at lower bitrates, making it a crucial area for perception-oriented compression models to address.

## 2.3 VIDEO VARIATIONAL AUTOENCODER

Video generation has received significant attention, with video latent diffusion models emerging as a key technology. These methods Brooks et al. (2024); Polyak et al. (2024); Yang et al. (2025) leverage pre-trained video VAEs to construct a perception-oriented latent space, followed by diffusion models to learn the distribution of latent variables. Consequently, the generative performance is fundamentally dependent on the reconstruction ability of the video VAE. As a result, increasing attention is being directed toward the development and optimization of video VAEs.

Video VAE processes a group of pictures simultaneously, eliminating both temporal and spatial redundancy for dimensionality reduction. Unlike image VAE, which focuses solely on spatial redundancy, video VAE also addresses temporal redundancy, making it ideal for video compression. Some approaches (Chen et al., 2024b; Zhao et al., 2024; Yang et al., 2025; Cheng & Yuan, 2025) employ dense $3D$ structures for high-quality compression, while others (Polyak et al., 2024; Peng et al., 2025) adopt a $2 + 1D$ architecture to balance redundancy reduction and computational efficiency, though at the cost of lower reconstruction quality. Video VAEs are typically trained with perceptual loss to map video clips into a perception-oriented latent space, enhancing visual qual-

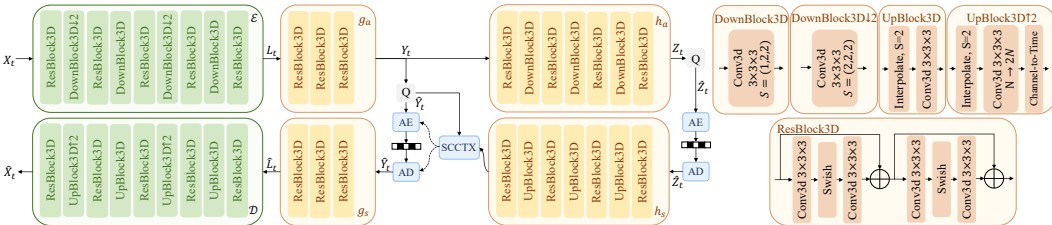

Figure 4: Architecture of our framework. $\uparrow$ and $\downarrow$ respectively denote upsampling and downsampling along the temporal dimension. $Conv3d$ $M \times M \times M$ represents a $3D$ convolution with a kernel size of $M$. $S = (T, H, W)$ indicates the stride along the temporal, height, and width dimension. $N \to M$ denotes a channel transformation from $N$ to $M$ channels.

ity while reducing dimensionality and computational complexity. This paper proposes a generative video compression framework based on video VAE, efficiently representing video data and compressing latent variables to achieve superior perceptual quality at low bitrates.

## 3 PROPOSED METHOD

In this section, we present the details of the proposed method, as illustrated in Figure 3. To achieve perceptual video compression at low bitrates, we employ a video VAE to encode images into a perceptually optimized latent space and compress the latent variables using a hierarchical architecture, further capturing spatial redundancy, thereby reducing the bitrate. Let $\mathbf{X} = \{X_1, X_2, \ldots, X_{t-1}, X_t, \ldots\}$ denote a video, where $X_t = \{x_{t,1} \ldots x_{t,T}\} \in \mathbb{R}^{T \times H \times W \times 3}$ represents a group of $T$ original video frames, with each frame $x_{t,i} \in \mathbb{R}^{H \times W \times 3}$ in RGB format. As shown in Figure 3, a group of pictures $X_t$ is first encoded into a latent variable $L_t \in \mathbb{R}^{t \times h \times w \times c}$ using the encoder $\mathcal{E}$ of the video VAE. The latent $L_t$ is then transformed by an analysis transform $g_a$ into a latent representation $Y_t$, which further eliminates spatial redundancy. Next, $Y_t$ is quantized to $\hat{Y}_t$ and losslessly compressed using entropy coding. During decoding, the synthesis transform $g_s$ reconstructs the latent variable $\hat{L}_t$, which is then processed by the decoder $\mathcal{D}$ of the video VAE to obtain the reconstructed image set $\hat{X}_t$. Since different GoPs are encoded and decoded independently, our method naturally supports parallel processing, enabling efficient acceleration. To further improve visual consistency across GoPs, we introduce a frame-overlapping strategy that enhances temporal continuity without requiring additional supervision.

### 3.1 VIDEO VARIATIONAL AUTOENCODER

We employ a causal video VAE with a dense $3D$ architecture to ensure the reconstruction quality, as illustrated in Figure 4. The video VAE downsamples input video clip $X_t \in \mathbb{R}^{T \times H \times W \times 3}$ with a temporal scale of $\rho_t = \frac{T-1}{t-1}$ and a spatial scale of $\rho_s = \frac{H}{h} = \frac{W}{w}$. This process transforms the input GoP $X_t$ into latent variable $L_t \in \mathbb{R}^{t \times h \times w \times c}$, where $t = 1 + \frac{T-1}{\rho_t}$ and $c$ represents the output channels of the encoder $\mathcal{E}$. The encoding and decoding processes are then defined as $L_t = \mathcal{E}(X_t)$ and $\tilde{X}_t = \mathcal{D}(L_t)$, where $\tilde{X}_t$ denotes the reconstructed video clip obtained from latent variable $L_t$.

In video generation, the common setting for $c$ is relatively small; however, this configuration presents significant drawbacks for video compression, as a small $c$ leads to insufficient information retention and poor reconstruction quality. To address this, we adjust the number of channels $c$ in the video VAE and fine-tune it specifically for video compression. To compress the video clip into a perceptually optimized latent space, we train the video VAE using the following perceptual loss function, which measures the discrepancy between the reconstructed clip $\tilde{X}_t$ and the original $X_t$.

$$\mathcal{L}_{VAE} = \alpha \, ||X_t - \tilde{X}_t||_2^2 + \beta \, \mathcal{P}(X_t, \tilde{X}_t)$$
$$= \frac{1}{T} \sum_{i=1,\ldots,T} \left( \alpha \, ||x_{t,i} - \tilde{x}_{t,i}||_2^2 + \beta \, \mathcal{P}(x_{t,i}, \tilde{x}_{t,i}) \right) \quad (1)$$

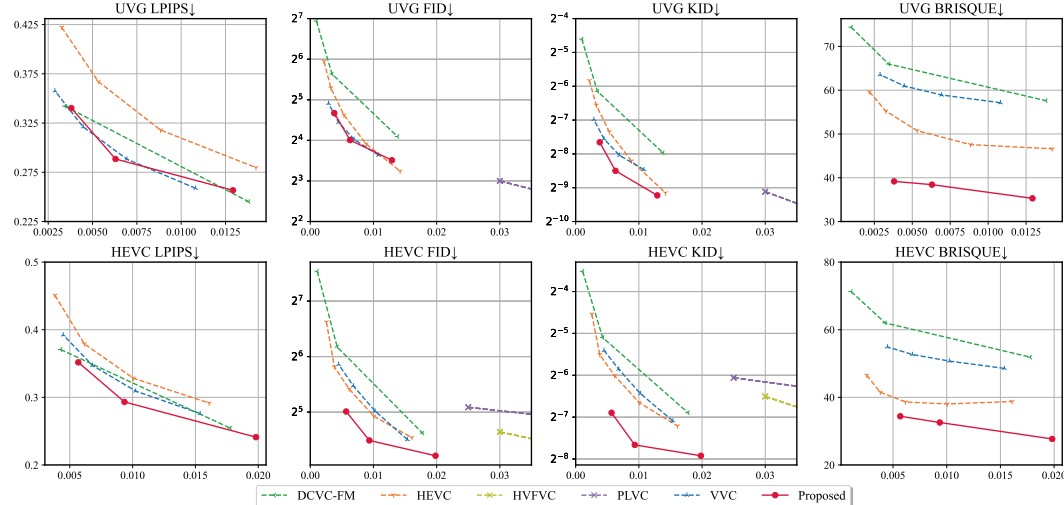

Figure 5: Numerical comparison results on the UVG, and HEVC datasets. The horizontal axis represents bits per pixel (bpp), and ↓ indicates lower values correspond to better performance. Results on the MCL-JCV dataset are provided in the supplementary.

where $\mathcal{P}$ represents a perceptual metric such as LPIPS (Zhang et al., 2018), and $\alpha, \beta$ are trade-off parameters balancing distortion and perceptual quality. The loss function computes the average distortion across frames in the video clip. Although video VAE applies downsampling, achieving a compression ratio of $\frac{3 \times \rho_t \times \rho_s^2}{c}$, the number of latent channels $c$ is typically larger than the original three RGB channels, resulting in an insufficient compression ratio. To further capture redundancy and improve the compression ratio, we leverage methods from image compression, as our architecture shares similarities with theirs.

## 3.2 HIERARCHICAL CODING OF VAE LATENTS

Given the trained video VAE and its perceptually optimized latent space, we further apply transform coding to capture spatial redundancy, enabling a significantly higher compression ratio. The latent variable $L_t \in \mathbb{R}^{t \times h \times w \times c}$ is processed by the analysis transform $g_a$, producing a more compact latent representation $Y_t \in \mathbb{R}^{t \times h \times w \times c}$. To effectively model spatial dependencies in $Y_t$, we draw inspiration from techniques widely adopted in image compression. Specifically, we incorporate a hyperprior (Ballé et al., 2018) architecture combined with a space-channel context model (He et al., 2022). This allows us to leverage an entropy model, such as arithmetic coding, to efficiently compress the quantized representation $\hat{Y}_t$ into a bitstream for storage and transmission.

To train the model efficiently, we first freeze the video VAE components $\mathcal{E}$ and $\mathcal{D}$ and train only the compression module to encode the latent variable $L_t$. Finally, we fine-tune the entire framework to achieve better compression performance. Since our goal is to optimize perceptual quality in the image domain, both training stages are supervised in the pixel space using the following loss function.

$$\begin{aligned}
\mathcal{L} &= \lambda \left( \alpha \, ||X_t - \hat{X}_t||_2^2 + \beta \, \mathcal{P}(X_t, \hat{X}_t) \right) + \mathcal{R}(\hat{Y}_t) + \mathcal{R}(\hat{Z}_t) \\
&= \frac{1}{T} \sum_i \left[ \lambda \left( \alpha \, ||x_{t,i} - \hat{x}_{t,i}||_2^2 + \beta \, \mathcal{P}(x_{t,i}, \hat{x}_{t,i}) \right) + \mathcal{R}(\hat{y}_{t,i}) + \mathcal{R}(\hat{z}_{t,i}) \right]
\end{aligned} \tag{2}$$

where $\mathcal{R}(\cdot)$ represents the estimated bitrate of the quantized latent representation, and $\lambda$ controls the trade-off between rate and perception.

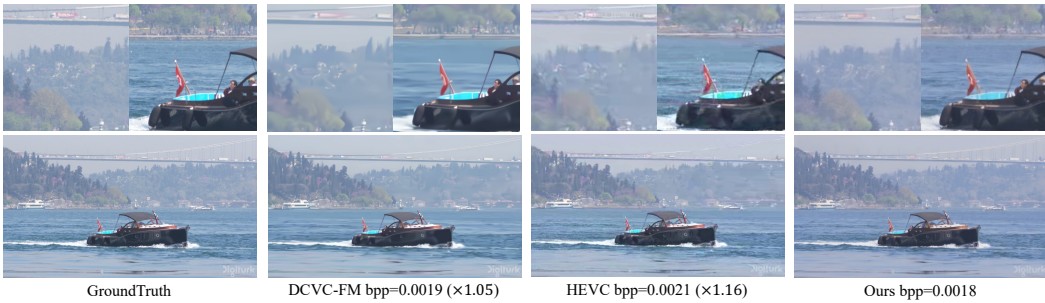

GroundTruth  DCVC-FM bpp=0.0019 (×1.05)  HEVC bpp=0.0021 (×1.16)  Ours bpp=0.0018

(a) Reconstructed frame 201 from Bosphorus sequence in the UVG dataset.

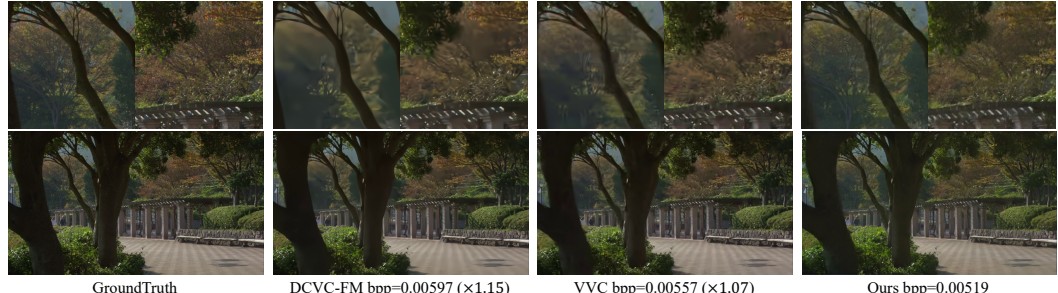

GroundTruth  DCVC-FM bpp=0.00597 (×1.15)  VVC bpp=0.00557 (×1.07)  Ours bpp=0.00519

(b) Reconstructed frame 21 from ParkScene sequence in the HEVC B Test Set.

Figure 6: Visual comparison of different methods. See supplementary for more results.

### 3.3 TRAINING-FREE TEMPORAL CONTINUITY ENHANCEMENT

While the above process ensures the perceptual quality of individual frames and maintains temporal continuity within a GoP, it processes each video clip independently, and the temporal correlations between adjacent GoPs are not explicitly modeled. This may result in noticeable discontinuities, which may be further exacerbated by quantization.

To address this issue, we propose a frame-overlapping strategy that requires no additional training to enforce the temporal consistency between GoPs. Specifically, we introduce overlap between adjacent GoPs by letting them share a transition frame. For two adjacent GoPs $X_t = \{x_{t,1}, ..., x_{t,T}\}$ and $X_{t+1} = \{x_{t+1,1}, ..., x_{t+1,T}\}$, the overlapping frame is $x_{t,T} = x_{t+1,1}$. During decoding, the reconstructed frames $\hat{x}_{t,T}$ and $\hat{x}_{t+1,1}$ are fused to generate a seamless transition frame through weighted interpolation. $\hat{x}_{t,T} = \hat{x}_{t+1,1} = \theta \, \hat{x}_{t,T} + (1 - \theta) \, \hat{x}_{t+1,1}$, where $\theta$ is a blending factor that balances the contribution of both frames, ensuring smooth transitions between GoPs.

Although this approach introduces a slight increase in bitrate due to redundant encoding, it effectively maintains continuity between different GoPs without requiring additional training. For a video with $N \times T$ frames, the frame-overlapping strategy results in the redundant encoding of $\frac{N \times T}{T-1} - 1$ frames. Consequently, the total bitrate increases by a factor of $\frac{1}{T-1} - \frac{1}{N \times T}$. In low-bitrate scenarios, the additional overhead remains minimal, with the increase in bpp not exceeding $3 \times 10^{-3}$, which is negligible.

## 4 EXPERIMENTS

**Training Details.** For training, we utilize $12,844$ videos from the high-quality long-video dataset OpenVid (Nan et al., 2025) to ensure both frame quality and sufficient video length. Frames are extracted every third frame, and each clip is randomly cropped to a resolution of $256 \times 256$. Our video VAE is primarily based on CV-VAE (Zhao et al., 2024), with modifications to the number of channels to better align with the characteristics of compression. We train the framework using the AdamW optimizer with a batch size of 4 and a learning rate of $5.0 \times 10^{-6}$. To balance

Table 1: Quantized temporal continuity results on the HEVC test set using tOF↓ and tLP↓.

| BPP(Before) | 0.0198 | 0.0093 | 0.0056 |
|---|---|---|---|
| tOF/tLP(Before) | 0.81/0.096 | 1.10/0.10 | 1.10/0.10 |
| BPP(After) | 0.0217 | 0.0102 | 0.0061 |
| tOF/tLP(After) | 0.72/0.086 | 0.909/0.088 | 0.98/0.091 |

distortion and perception, we set $\alpha = 10^{-3}, \beta = 1$, and to accommodate variable rates, we use $\lambda \in \{0.2, 0.1, 0.05\}$. Other default settings include $c = 128, \rho_t = 4, \rho_s = 8, T = 9, \theta = 0.5$.

**Evaluation.** We evaluate our method using both distortion and perceptual quality metrics, conducting all evaluations on full-resolution videos. Bit per pixel (bpp) is used to measure the average number of bits required to encode a single pixel in each frame. For video quality assessment, we select several widely used metrics. To quantify both distortion and perceptual quality, we employ PSNR, LPIPS (Zhang et al., 2018), Fréchet Inception Distance (FID) (Heusel et al., 2017), Kernel Inception Distance (KID) (Binkowski et al., 2018), and BRISQUE (Mittal et al., 2011). Among these, PSNR and LPIPS are full-reference metrics, while the others are no-reference metrics. We evaluate our method alongside baselines on the HEVC test set (Flynn et al., 2011), UVG (Mercat et al., 2020), and MCL-JCV (Wang et al., 2016).

**Baseline.** We compare our method with several representative and state-of-the-art (SOTA) video compression approaches. For the widely-used traditional codec HEVC (Sullivan et al., 2012), we use FFmpeg with the default mode, following (Hu et al., 2021). For the SOTA traditional method, we also compare against VVC (Wieckowski et al., 2021) and follow the command line settings from (Li et al., 2024b). For distortion-oriented neural codecs, we evaluate against the advanced DCVC-FM (Li et al., 2024b). For perception-oriented approaches, we compare with PLVC (Yang et al., 2022) and HVFVC (Li et al., 2023b). For methods with publicly available codes and models, we reproduce the results to ensure consistency in testing and evaluation. For methods without released models, we report the metrics provided in their paper for comparison.

**Main Results.** To quantify our performance, we adopt widely used metrics for evaluation. While LPIPS measures similarity between individual image pairs, FID and KID assess the distribution similarity between the references and reconstructions. BRISQUE evaluates image quality based on natural image statistics. The Rate-Perception curves for HEVC, VVC, DCVC-FM, PLVC, HVFVC, and ours are shown in Figure 5. Our approach consistently outperforms others in terms of perceptual quality metrics. Although HVFVC and PLVC achieve similar FID and KID scores to ours, they require more than twice the bitrate to reach the same level of perceptual quality. On the HEVC test set, our method reduces bitrate by 21.59% compared to VVC at the same LPIPS level, and achieves a 52.53% bitrate saving at the same FID level. While VVC achieves comparable LPIPS scores in other datasets, our method surpasses it in non-reference metrics. This indicates that VVC prioritizes reconstruction fidelity, leading to strong full-reference metric performance. In contrast, our method focuses on perceptual generation, which, though less pixel-consistent, produces visually appealing results that are difficult to distinguish from the original frames. These results demonstrate the effectiveness of our approach in achieving high visual quality.

**Visual Results.** The visual results are presented in Figure 6, where we compare reconstructions of various methods on different textures, including trees, cars, and lakes. To highlight advantages of our approach, we set the bitrates of HEVC, VVC, and DCVC-FM slightly higher than ours. As shown in Figure 6, ours achieves richer and more photo-realistic textures at lower bitrates than other approaches. For example, in Figure 6a, HEVC and DCVC-FM suffer from motion artifacts on the cars, leading to motion tails and degraded perceptual quality, whereas our method preserves fine details and accurately reconstructs car contours. Similarly, in Figure 6b, VVC and DCVC-FM produce blurry trees and leaves, while our model restores clearer, more detailed textures that closely match the original. In general, our method achieves more realistic and visually appealing results by effectively reducing artifacts and blurriness.

**Time Continuity.** To quantitatively evaluate temporal consistency, we report tOF and tLP (Chu et al., 2020; Oh & Kim, 2022; Youk et al., 2024) in Table 1, reflecting the effectiveness of our temporal continuity strategy. tOF measures the pixel-wise difference of estimated motions, defined as $tOF = ||OF(\hat{x}_{t-1}, \hat{x}_t) - OF(x_{t-1}, x_t)||_1$, where $OF$ represents optical flow estimation using

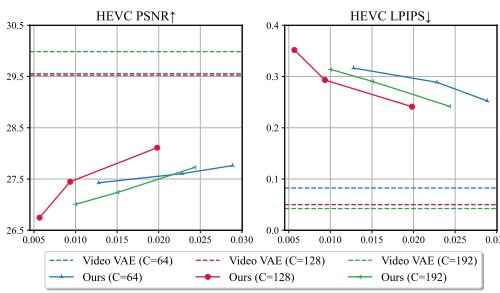

Figure 7: Ablation study of channel numbers $c$.

Table 2: Encoding/decoding speed (fps) and FID-BDRate.

| Method | Enc. fps | Dec. fps | FID-BDRate |
|---|---|---|---|
| VTM-17.0 | 0.01 | 23.6 | -27.24% |
| DCVC-DC | 3.3 | 4.3 | - |
| DCVC-FM | 5.0 | 5.9 | 0.00% |
| CV-VAE | 3.18 | 1.53 | - |
| Ours (CV-VAE) | 3.04 | 1.48 | -64.81% |
| LeanVAE | 2425 | 44.01 | - |
| Ours (LeanVAE) | 919 | 24.04 | -39.54% |

DIS (Kroeger et al., 2016). tLP evaluates perceptual changes over time using deep feature maps, defined as $tLP = ||LP(\hat{x}_{t-1}, \hat{x}_t) - LP(x_{t-1}, x_t)||_1$, where $LP$ refers to the perceptual LPIPS metric (Zhang et al., 2018). Our results demonstrate that the proposed frame-overlapping strategy introduces only a small bitrate overhead while significantly improving temporal consistency. Furthermore, we observe that this strategy provides greater benefits at higher bitrates. This is because, at higher bitrates, intra-GoP temporal consistency is already well preserved, making inter-GoP discontinuities the primary source of inconsistency. By addressing these inter-GoP discontinuities, our approach further enhances overall temporal smoothness.

**Number of Channels.** The output channel $c$ of the video VAE is a key hyperparameter that significantly impacts overall performance (Yao et al., 2025). The compression ratio of video VAE, given by $\frac{3 \times \rho_t \times \rho_s^2}{c}$, is primarily influenced by $c$, as $\rho_s = 8$, $\rho_t = 4$ are widely used in video VAE. Although $c = 16$ is commonly adopted in video generation, it is suboptimal for video compression as it fails to preserve important texture details. A smaller $c$ results in greater information loss but enables a higher compression ratio, leading to a lower-dimensional latent space that may be advantageous for compression. Conversely, a larger $c$ reduces the compression ratio but retains more information, potentially improving visual quality. As shown in Figure 7, video VAEs with $c = 64, 128, 192$ achieve compression ratios of $12, 6, 4$, respectively. After entropy coding, all configurations yield a final compression ratio exceeding $800$, with $c = 128$ providing the best result in both PSNR and LPIPS. This is because $c = 128$ provides an optimal trade-off between preserving information and maintaining an efficient compression ratio. Therefore, we set $c = 128$ as our default configuration.

**Complexity Analysis.** Table 2 presents the encoding and decoding speeds, along with FID-BDRate comparisons. The encoding speed of our method with CV-VAE, denoted as Ours(CV-VAE), is comparable to DCVC-DC. In contrast, the decoding process is relatively slower due to the use of upsampling and an auto-regressive context model. Despite this, Ours(CV-VAE) achieves a significantly better FID-BDRate of -64.81% compared to DCVC-FM. GoP-level parallelization further improves efficiency even on a single GPU, where running 4 batches simultaneously increases encoding speed by 27% and decoding speed by 23%. Notably, the majority of computation in Ours(CV-VAE) is attributed to the CV-VAE backbone, while latent-domain compression accounts for less than 5% of the total runtime. To further demonstrate the efficiency of our framework, we replace the backbone with LeanVAE (Cheng & Yuan, 2025), a highly efficient video VAE. The resulting model, Ours(LeanVAE), supports real-time decoding at 24 fps while still achieving a substantial FID-BDRate improvement of $-39.54\%$. These results highlight the framework's compatibility with lightweight backbones for real-time applications.

## 5 CONCLUSION

This paper presents a novel neural video compression framework based on video variational autoencoders. The framework operates on groups of frames, leveraging the video VAE to encode video clips into a perceptually optimized latent space. A hyperprior-based architecture is designed to capture redundancy more effectively. We propose an overlapping strategy to preserve temporal consistency across groups. Comprehensive experimental results validate the superior perceptual performance of the proposed framework, outperforming both traditional video coding standards and learning-based video compression methods.

ETHICS STATEMENT

All the authors read and adhere to the ICLR Code of Ethics.

REPRODUCIBILITY STATEMENT

We ensure that all experiments in this paper are fully reproducible as described in the main text.

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

# A ARCHITECTURE

Figure 4 illustrates the detailed architecture of our framework when using CV-VAE Zhao et al. (2024) as the backbone. When LeanVAE Cheng & Yuan (2025) is used instead, only the structures of the video VAE's encoder $\mathcal{E}$ and decoder $\mathcal{D}$ are replaced with their simplified counterparts. For simplicity and fair comparison, all other components of the network remain unchanged. To ensure effective information preservation, we set the output channel dimensions of $\mathcal{E}$, $g_a$, and $h_a$, that is, $L_t$, $Y_t$, and $Z_t$, to the same value $c$.

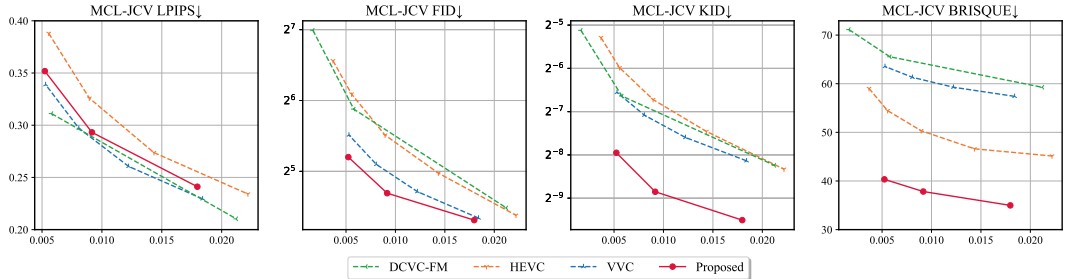

Figure 8: Numerical comparison results on the MCL-JCV dataset.

# B MORE QUANTITATIVE RESULTS

The Rate-Perception curves on the MCL-JCV dataset Wang et al. (2016) for HEVC Sullivan et al. (2012), VVC Bross et al. (2021), DCVC-FM Li et al. (2024b), and our method are shown in Figure 8. Although our LPIPS scores on MCL-JCV are not particularly strong, we achieve significantly better results in FID, KID, and BRISQUE, indicating superior perceptual quality in the reconstructed videos. This observation is consistent with the qualitative results we visualize.

# C MORE VISUAL RESULTS

Additional visual results are presented in Figure 9, where we compare the reconstructions across various textures, including roofs, trees, faces, hoofs, and buildings. To highlight the strengths of our approach, the bitrates of HEVC Sullivan et al. (2012), VVC Bross et al. (2021), and DCVC-FM Li et al. (2024b) are set slightly higher than ours. As shown, our method consistently produces richer and more photo-realistic textures at lower bitrates. For instance, in Figure 9a, reconstructions from DCVC-FM and VVC exhibit noticeable blurriness in roofs and leaves, while our approach retains clearer and more detailed structures. As illustrated in Figure 9b, both HEVC and VVC reconstructions suffer from motion artifacts on the hooves and fences, leading to a loss of texture details and reduced perceptual quality. In contrast, our method maintains fine details and achieves a higher overall reconstruction quality. In Figure 9c, our reconstruction avoids motion artifacts and retains the texture of the trousers. Additionally, in Figure 9d, DCVC-FM suffers from blurriness and HEVC exhibits artifacts, while our method produces much better reconstructions with fine details. Overall, these results confirm that our approach effectively reduces blurriness and motion artifacts, delivering more realistic and visually pleasing reconstructions.

# D VIDEO DEMOS

We present an extended visual comparison between our method (left) and the widely-used traditional codec HEVC Sullivan et al. (2012) (right) on a video sequence from the MCL-JCV Wang et al. (2016) dataset. The bitrates are annotated in the top-left and top-right corners for clear reference. Although HEVC operates at a slightly higher bitrate, it introduces noticeable blocking and blurring artifacts. In contrast, our method delivers significantly better visual quality, preserving finer details and textures. For clearer observation, we also provide two short video clips, which further highlight the perceptual advantages of our approach over HEVC. To facilitate visual inspection, we slow down the playback speed of the videos by reducing the frame rate to half of the original.

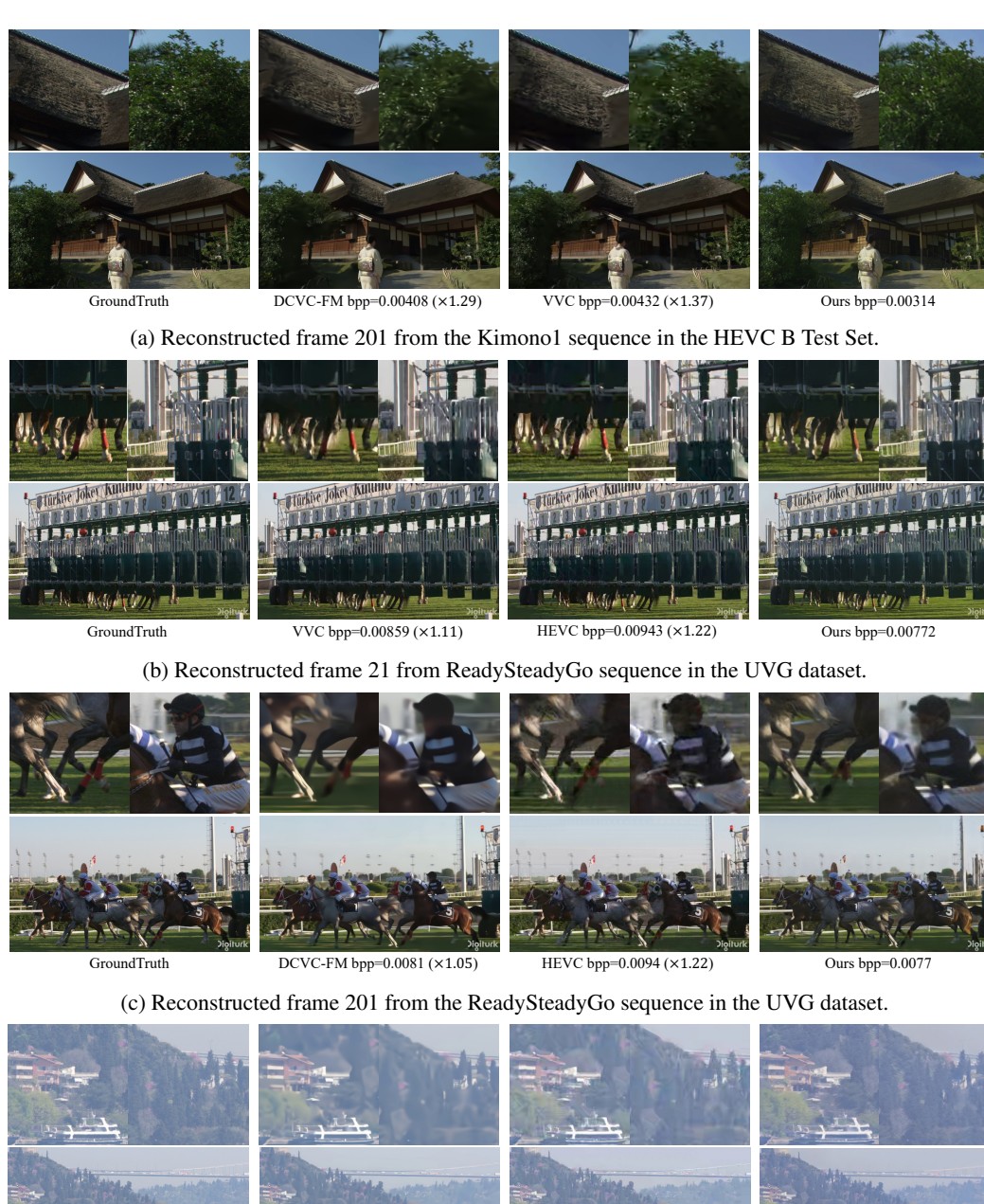

(a) Reconstructed frame 201 from the Kimono1 sequence in the HEVC B Test Set.

(b) Reconstructed frame 21 from ReadySteadyGo sequence in the UVG dataset.

(c) Reconstructed frame 201 from the ReadySteadyGo sequence in the UVG dataset.

(d) Reconstructed frame 21 from the Bosphorus sequence in the UVG dataset.

Figure 9: Visual comparison of different methods.

# E   THE USE OF LARGE LANGUAGE MODELS (LLMS)

This paper uses a large language model solely to assist with minor language polishing and grammar refinement. All research ideas, experimental designs, analyses, and conclusions are developed entirely by the authors.

