# OpenReview forum: "Perceptual Neural Video Compression with Video Variational AutoEncoder at Low Bitrates"
_ICLR.cc/2026/Conference — Submitted to ICLR 2026_

### Official Review · Reviewer_8vdS · 2025-10-29

**Soundness:** 2
**Presentation:** 2
**Contribution:** 2
**Rating:** 4
**Confidence:** 5

**Summary:**

The paper proposes a perceptual video codec that encodes entire GoPs with a pretrained Video-VAE (CV-VAE or LeanVAE) to get a perceptually friendly latent, then apply a hyperprior entropy model in latent space, with no explicit motion estimation/compensation modules. A training-free GoP-overlap (one shared transition frame with weighted fusion) improves inter-GoP continuity at tiny bitrate overhead. Results show perceptual gains vs HEVC/VVC/DCVC-FM on LPIPS/FID/KID/BRISQUE, and a LeanVAE variant reaches ~24 fps decode, while keeping a sizable FID-BDRate improvement.

**Strengths:**

- The system is clearly presented, empirically supported, and well-motivated for the low-bitrate perceptual regime. The overlap trick is simple and practical, and the ablations are thorough. The authors include ablations on latent channel width (c = 64, 128, 192), overlap weight $\theta$, and different perceptual losses, showing how each impacts perceptual and temporal metrics. These details demonstrate careful experimentation and help readers reproduce the setup.
- Quantitatively, the codec achieves consistent gains in FID-BDRate and LPIPS compared to HEVC, VVC, and DCVC-FM, particularly below 0.05 bpp. Qualitatively, reconstructions exhibit fewer blocking and temporal flicker artifacts, suggesting that the VAE prior effectively captures smooth temporal evolution.
- The proposed training-free overlap, which blends one frame between neighboring GoPs, helps mitigate the temporal discontinuities that often plague GoP-based codecs. The idea is lightweight, easy to implement, and incurs almost no bitrate overhead while yielding measurable improvements on tOF/tLP.

**Weaknesses:**

- **Limited novelty.**: In my opinion, neither the idea of non-explicit motion estimation nor the use of video generation-style (3D) autoencoders / tokenizers are novel in the video compression community. For example, MAGVIT-v2 [1] (though not specifically designed for video compression but did validate for perceptual compression performance) and the more recent DHVC [2] and GiVIC [3] have both adopted 3D autoencoder (and 3D autoencoder only) and patchification to consume a group/chunk of frames for joint spatiotemporal redundancy reduction. The same goes for hierarchical entropy coding or the utilization of spatial-channel context [4]. Though I think the training-free temporal continuity strategy is an interesting addition, the differences and novelties for the other two major contributions should be more clearly discussed in the paper.
- **Evaluation scope**. While the results convincingly show perceptual gains at very low bitrates (and the model is designed targeting low bitrates more), the paper does not explore how the proposed method performs at moderate or high bitrates, where pixel fidelity becomes more relevant. It would strengthen the work if the authors could report whether the video VAE remains competitive in those regimes or if its advantage diminishes once distortion metrics dominate.
- **Complexity profiling.** The claimed real-time decoding is achieved only with LeanVAE; the heavier CV-VAE version remains far from real-time. A more detailed latency breakdown (tokenization, I/O, entropy, AR context, upsampling) would help clarify bottlenecks and show whether the method scales to higher resolutions.

Overall, I think the paper is intuitive and of good quality, but the claim "eliminating the need for motion estimation and compensation" seems incremental or even not entirely novel to me. It would be helpful if the authors are able to demonstrate why such a video VAE is particularly preferable in the case of perceptual-oriented compression, significantly different from some existing methods, or is able to yield superior compression performance when optimized for distortion-oriented losses.

[1] Language Model Beats Diffusion - Tokenizer is key to visual generation, ICLR'24

[2] Deep Hierarchical Video Compression, AAAI'24

[3] GIViC: Generative Implicit Video Compression, ICCV'25

[4] Neural Video Compression With Diverse Contexts, CVPR'23

**Questions:**

Please refer to the **Weaknesses** section.

---

### Official Review · Reviewer_FWxT · 2025-10-30

**Soundness:** 2
**Presentation:** 2
**Contribution:** 2
**Rating:** 2
**Confidence:** 4

**Summary:**

This paper proposes a neural video compression framework that leverages a Video Variational AutoEncoder (Video VAE) to jointly encode an entire Group of Pictures (GoP) into a perceptual latent space, rather than relying on the conventional motion estimation and compensation pipeline used in traditional or neural codecs. To encourage temporal smoothness across GoPs, the method introduces a training-free frame-overlapping strategy between adjacent GoPs. Experiments on the HEVC, UVG, and MCL-JCV datasets show improvements in perceptual metrics such as FID, KID, and BRISQUE at low bitrates (<0.03 bpp), compared with HEVC, VVC, and DCVC-FM.

**Strengths:**

1. The paper leverages the strong spatio-temporal joint modeling capability of Video VAE. By jointly encoding entire GoPs, the method in principle avoids the need for explicit motion estimation and motion compensation, and claims to effectively eliminate both spatial and temporal redundancy.
2 .The proposed training-free overlapping coding strategy that smooths transitions between GoPs is easy to implement and incurs only a small bitrate overhead under low-bitrate scenarios. The paper also provides quantified temporal continuity metrics (tOF and tLP) comparing results before and after overlapping.

**Weaknesses:**

1. Although the paper claims to address perceptual degradation at lower bitrates (below 0.03 bpp), the experimental section lacks results at bitrates near 0.03bpp. Moreover, evaluations on the UVG, HEVC, and MCL-JCV datasets indicate that the proposed method does not exhibit clear advantages over other methods in terms of LPIPS.
2. Some of the baseline methods used for comparison (e.g., VVC and DCVC-FM) are relatively outdated and were primarily optimized for rate–distortion (RD) performance rather than perceptual quality. This makes the comparison potentially less fair or conclusive.
3. The paper lacks an ablation study on the blending factor θ and the number of overlapping frames, which would help evaluate the sensitivity of the method to these parameters.

**Questions:**

1. How sensitive is the method to the choice of the pretrained Video VAE, and have alternative VAE backbones been tested for generalization? To what extent does the chosen VAE architecture contribute to the final compression performance?
2. Does the training strategy significantly impact both convergence speed and perceptual performance? How was the final training strategy selected and experimentally validated?

---

### Official Review · Reviewer_zWCo · 2025-10-30

**Soundness:** 3
**Presentation:** 3
**Contribution:** 3
**Rating:** 6
**Confidence:** 4

**Summary:**

This paper presents a framework for perceptual neural video compression based on a video variational autoencoder (VAE) operating at low bitrates. The proposed system consists of three main components: a generative video VAE encoder, a transform block that compresses the encoder’s output, and a hyperprior module for entropy coding. The paper is clearly written and the experimental results are promising; however, the approach offers limited technical novelty. Additionally, some aspects remain unclear and should be clarified by the authors.

**Strengths:**

1. The paper is well written and clearly presented.

2. The experimental results are solid and support the proposed approach.

3. The work can serve as a valuable reference point for future research in the field.

**Weaknesses:**

1. Some points in the paper are unclear (see questions).

2. A few formulas appear to be incorrect (see questions).

3. The paper lacks a PSNR comparison with state-of-the-art methods.

**Questions:**

1. The size of $L_t$ is unclear. The authors mention $t$ as the first dimension size, but this cannot be correct, since $t$ is simply the index of the clip. Later in the paper (Section 3.1), a definition of $t$ is provided, but it remains ambiguous, as $t$ was previously defined as the clip index. Do the authors actually mean that the size of the latent representation depends on the clip number? This point requires clarification.

2. Is the size of $Y_t$ really the same as that of $L_t$? This seems rather unusual. Similarly, why are $g_a$ and $g_s$ models so much smaller than the hyperprior models? This choice is not very intuitive.

3. Regarding Fig. 7, I do not understand why the same number of channels is maintained for $L$ and $Y$. From the dashed lines, it appears that the number of channels of $L$ has little impact on performance, whereas the number of channels of $Y$ seems to be the crucial factor.

4. The authors should report the performance in terms of PSNR, not just perceptual metrics. This is important to assess how accurately the reconstruction can diverge from the original uncompressed video.

5. You wrote, "where $P$ represents a perceptual metric such as LPIPS." Does this mean that the results in Figure 5 were obtained by training a separate model for each metric? In other words, did you train one model for LPIPS, another for FID, and so on?

6. (Minor) I would suggest testing more than 3 rate points.

---

### Official Review · Reviewer_apEs · 2025-11-01

**Soundness:** 2
**Presentation:** 2
**Contribution:** 2
**Rating:** 2
**Confidence:** 5

**Summary:**

This paper introduces a 3D Video VAE-based video compression framework, which eliminates the need for motion estimation and compensation. Instead, the model leverages video VAE to jointly capture spatial and temporal dependencies. To ensure temporal consistency between different GoPs, the authors further propose a simple overlapping processing strategy, allowing smooth transitions between adjacent frame groups

**Strengths:**

The idea of using 3D VAE for direct video compression is interesting.

**Weaknesses:**

- This work appears somewhat incremental, representing a straightforward extension of 2D variational autoencoder–based image codecs to 3D variational autoencoder–based video codecs, without introducing any novel modules. Fundamental concepts such as hierarchical coding and the entropy model are not new. Therefore, this should not be considered a primary contribution of the paper.

- This work lacks a comprehensive comparison of perceptual quality, and additional metrics should be included for a fair evaluation. Moreover, since LPIPS is a strong indicator of perceptual quality, the method fails to outperform the baselines.

- The proposed group-level processing may function similarly to B-frame coding, potentially offering an advantage not present in other baseline methods.

- Furthermore, the applicability of this work is limited, as the model processes groups of nine frames at a time, potentially introducing significant delay and restricting its use in real-time or low-latency scenarios.

- BD-rate evaluation is performed using only three bitrate points, which may not yield reliable or smooth curves for comparison.

**Questions:**

- Although fidelity is not the primary focus of this work, a PSNR comparison with the baselines should be provided to ensure the method maintains high reconstruction quality.

- Additionally, results using perceptual quality metrics, such as DISTS, should be reported.

---

### Official Review · Reviewer_R79P · 2025-11-02

**Soundness:** 3
**Presentation:** 3
**Contribution:** 3
**Rating:** 6
**Confidence:** 5

**Summary:**

This paper proposes a perceptual neural video compression framework leveraging video Variational AutoEncoders (VAEs) to achieve high perceptual quality at low bitrates. Instead of operating on a frame-by-frame basis, the approach encodes entire groups of frames (GoPs) to jointly eliminate spatial and temporal redundancies. The compressed latent representation is further optimized using transform and entropy coding, while an overlapping GoP strategy maintains temporal consistency between video groups. Extensive experiments on HEVC, UVG, and MCL-JCV benchmarks demonstrate improvement over both classical (HEVC, VVC) and state-of-the-art neural codecs in perceptual metrics, particularly at low bitrates.

**Strengths:**

**Core conceptual originality:** The shift from frame-wise, motion-compensation paradigms to a holistic, group-of-pictures encoding grounded in video VAEs is meaningful and opens new design space for neural video compression (see Figure 2, which visually contrasts this group-based paradigm against prior methods).

**Strong empirical results:** The approach consistently outperforms both traditional and advanced learning-based codecs (HEVC, VVC, DCVC-FM, PLVC, HVFVC) on several perceptual quality benchmarks. For instance, Figure 5 and Table 2 exhibit clear quantitative improvements in FID, LPIPS, KID, and BRISQUE at low bitrates across multiple data sets.

**Solid ablation and hyperparameter tuning:** The impact of the latent channel size is presented clearly in Figure 7, with a thoughtful analysis on compression ratio and perceptual quality.

**Weaknesses:**

**Ablation, but incomplete sensitivity analysis:**  While the effect of the channel size $c$ is analyzed in Figure 7, other key hyperparameters (e.g., the GoP size, the overlapping window length, or the blending factor $\theta$) are only mentioned as fixed values. A more systematic ablation—e.g., does increasing the overlap length or adjusting the blending factor enhance temporal quality further, or does it merely increase redundancy? How robust is the method to varying GoP length?

**Perception-distortion tradeoff is underdeveloped:** Although perceptual quality is prioritized (as per FID, LPIPS, KID curves in Figure 5), the paper does not provide a deep exploration of the tradeoff behavior. For example, how does the perceptual gain compare to distortion loss in PSNR? And at what point does visual coherence break down (Section 4, Figure 5)? This is especially important given the negative impact on pixel-level fidelity for critical applications.

**Potential overstatement of generality:** Although the method is shown to outperform other codecs on standard datasets, its robustness to domains with rare events, streaming scenarios, or videos with complex, long-term motion is not explored. The parallelizable GoP-level coding could expose the system to unique artifacts in these cases.

**Questions:**

Please see Weaknesses.

---

### Meta-Review · Area_Chair_YpNd · 2026-01-01

**Summary:**

Reviewers generally agree that the paper is well written and that the experimental results are solid, showing perceptual improvements on various benchmarks at very low bitrates. However, multiple reviewers characterize the approach as offering limited technical novelty, viewing it as a straightforward extension of 2D VAE-based image codecs or existing 3D autoencoder-based video compression, without clear novel modules.

**Reviewer Concerns:**

The main concern across reviews is that the work is incremental. Reviewers note that key components -- like 3D VAEs, GoP-level processing, hierarchical entropy encoding, and the elimination of explicit motion estimation -- are not new. The paper lacks clear articulation of what fundamentally separate it from the existing methods.
- Several reviewers point out the evaluation is incomplete, PSNR is missing, LPIPS does not consistently outperform baselines, and additional perceptual metrics are requested.
- Reviewers also highlight insufficient ablation and sensitivity analysis for important design choices such as GoP size, overlap length, blending factor, and VAE backbone.
- Additional concerns include baseline fairness, limited applicability due to GoP-level latency, unclear notation or architectural choices

**Reviewer Scores:**

Reviewer scores range from 2 (reject) to 6 (marginally above the acceptance threshold). Authors didn't provide a rebuttal and so the scores from all reviewers stayed unchanged, with concerns about novelty, evaluation rigor and clarity left unaddressed. Overall, the reviews indicate a weak-reject consensus.

---

### Decision · Program_Chairs · 2026-01-26

Reject